# Is Abdominal Fat Distribution Associated with Chronotype in Adults Independently of Lifestyle Factors?

**DOI:** 10.3390/nu12030592

**Published:** 2020-02-25

**Authors:** Ramona De Amicis, Letizia Galasso, Alessandro Leone, Laila Vignati, Giulia De Carlo, Andrea Foppiani, Angela Montaruli, Eliana Roveda, Emiliano Cè, Fabio Esposito, Angelo Vanzulli, Alberto Battezzati, Simona Bertoli

**Affiliations:** 1International Center for the Assessment of Nutritional Status (ICANS), Department of Food Environmental and Nutritional Sciences (DeFENS), University of Milan, Via Sandro Botticelli 21, 20133 Milan, Italy; alessandro.leone1@unimi.it (A.L.); giulia.decarlo@unimi.it (G.D.C.); andrea.foppiani@unimi.it (A.F.); alberto.battezzati@unimi.it (A.B.); simona.bertoli@unimi.it (S.B.); 2Department of Biomedical Sciences for Health, University of Milan, Via Giuseppe Colombo 71, 20133 Milan, Italy; letizia.galasso@unimi.it (L.G.); angela.montaruli@unimi.it (A.M.); eliana.roveda@unimi.it (E.R.); emiliano.ce@unimi.it (E.C.); fabio.esposito@unimi.it (F.E.); 3IRCCS Istituto Ortopedico Galeazzi, Via Riccardo Galeazzi 4, 20161 Milan, Italy; 4Department of Oncology and Hemato-Oncology, University of Milan, 20122 Milan, Italy; angelo.vanzulli@unimi.it; 5IRCCS Istituto Auxologico Italiano, Obesity Unit and Laboratory of Nutrition and Obesity Research, Department of Endocrine and Metabolic Diseases, 20145 Milan, Italy

**Keywords:** chronotype, circadian typology, chrononutrition, abdominal obesity, visceral fat, abdominal fat distribution, Mediterranean diet

## Abstract

Both abdominal obesity and its visceral component are independently associated with cardiometabolic diseases. Among the non-modifiable and modifiable determinants, lifestyle plays a central role, while chronotype is an emerging factor. Evening type (E-Type), more active and efficient in the last part of the day, has been associated with a health-impairing style, resulting in a higher risk of obesity and cardiometabolic diseases than morning type (M-Type). However, no study has examined the contribution of chronotype to abdominal fat distribution, even considering adherence to the Mediterranean diet (MD). We conducted a cross-sectional study on 416 adults (69.5% females, 50 ± 13 years). Waist circumference (WC), visceral fat (VAT) using ultrasonography, chronotype through the reduced Morningness-Eveningness Questionnaire (rMEQ), and adherence to MD were studied. Our results showed no differences in WC and VAT between chronotypes. However, adherence to MD resulted significantly lower in the E-Types compared to M-Types. WC decreased with increasing Mediterranean score and rMEQ score, and VAT decreased with increasing rMEQ score, indicating that E-Types have +2 cm of WC and +0.5 cm of VAT compared to M-Types. In conclusion, these results showed that chronotype is independently associated with abdominal obesity and visceral fat, underlining the potential implications of the individual circadian typology on abdominal obesity.

## 1. Introduction

Abdominal obesity (measured as waist circumference [WC] ≥88 cm in women and ≥102 cm in men) is the primary factor in metabolic syndrome [1] independently of body mass index (BMI) [2,3]. Visceral abdominal (VAT) and subcutaneous abdominal fat tissue (SAT) [4] are the main components of abdominal fat. VAT is independently associated with insulin resistance, type 2 diabetes, abnormal lipid profile, hypertension, cardiovascular disease, colorectal cancer, and cognitive impairment [3,5,6], while the debate regarding the effect of subcutaneous adipose tissue (SAT) on metabolic risk is still open [4,6]. Widely accepted methods for measuring abdominal adipose tissue are computed tomography scans and magnetic resonance imaging [7]; however, due to their cost and exposure to radiation, they are not frequently used. Ultrasound is a non-invasive, reproducible, and reliable alternative representing, in addition to WC, the most commonly used surrogate measure of abdominal obesity on a clinical level [8].

Although some factors involved in the mechanisms underlying abdominal obesity are considered heritable [1], lifestyle determinants, such as diet and physical activity, remain the major driving forces [2]. It is known that a balanced diet, including the Mediterranean pattern (MD), is associated with a reduction of abdominal obesity, especially in the visceral compartment [2], particularly when combined with moderate- to high-intensity aerobic exercise. Indeed, the reduction of visceral adipose tissue in both overweight males and females is most likely due to these lifestyle factors [9], which significantly lower the risk of metabolic disorders [10], but only few studies examined whether a balanced diet together with physical activity mediates the interrelationships between chronotype and obesity in adults [11,12,13,14]. 

Among the risk factors of abdominal obesity, chronotype and circadian rhythms are emerging [15]. The interindividual differences observed in the natural propensity of behavioural manifestation relative to light-dark cycle are captured by a biological construct referred to as chronotype [16]. It can be considered as a continuum ranging between morning (M-Types) to evening chronotypes (E-Types), with M-Types who are prone to be more efficient and active during the first part of the day, and E-Types, who experience difficulties in getting up in the morning and need more time to reach the optimal level of both mental and physical efficiency [16]. Individuals lacking a marked circadian preference are referred to as intermediate- or neither-Types (N-Types), representing the most frequent circadian typology in the adult population [14]. 

Factors determining an individual’s chronotype are very complex, i.e., including both internal (e.g., cortisol, melatonin, and clock gene expression) and environmental (e.g., light-darkness cycle, seasons, cultural influences, social habits, urban lifestyle) factors [14,15]. Mounting literature suggests that the mismatch between light-dark cycle and social life, typical of E-Types, represents a risk factor for obesity and related cardiometabolic diseases [17,18]. Recently, the consequences of the discrepancy between social and biological time have been investigated and described as *social jet-lag*. In this perspective, it is possible to interpret the higher rates of sleep-related problems observed in E-types as a result of a daily social programme imposing a rhythm which is markedly misaligned with the biological one [19]. E-Types are mostly younger male adults, and they tend to have health-impairing behaviours [15,20], such as a lower adherence to a balanced diet, sedentary activity, smoking, and poorer sleep patterns, with respect to M-Types, [11,14,15], showing altered body composition and higher prevalence of metabolic disorders [12,13].

Although E-Types show a higher tendency to the consumption of noxious foods, such as sugary drinks, alcohol, and chocolate, rather than wholesome foods, such as fish, fruits, and vegetables [11,13,20,21], thus resulting in a lower diet quality according to the Baltic Sea Diet [11], Finnish researchers found that E-Types were no more susceptible to higher WC than other chronotypes [11]. On the other hand, Yu et al. [12], when examining the association between late chronotype and metabolic abnormalities, including visceral obesity, reported the independent correlation of E-Types with metabolic syndrome and diabetes, thus supporting the crucial role of circadian rhythms in the regulation of metabolism. Even in this research, VAT was not significantly associated with chronotype, but the contribution of diet was not investigated. Moreover, Zeròn-Rugerio et al. [22] observed that young adults with greater social jet lag and late chronotype displayed poorer adherence to the Mediterranean Diet (MD) and a higher Body Mass Index (BMI), but they only investigated general obesity and not abdominal fat distribution. In the light of these mixed results, the relation between chronotype and abdominal obesity and its compartments remain unclear.

In this scenario, we carried out a cross-sectional observational study to examine the contribution of chronotype to abdominal obesity and abdominal fat distribution in adults, considering sex, age, nutritional status, and adherence to MD.

## 2. Materials and Methods 

### 2.1. Study Design

We recruited 479 consecutive white European subjects who spontaneously attended the International Centre for the Assessment of Nutritional Status (ICANS) between April and December 2019 to be subjected to a structured nutritional assessment. The only inclusion criterion was age ≥ 18 years. Exclusion criteria were: (1) pregnancy and nursing; (2) significant cardiovascular, neurological, endocrine, and psychiatric disorders (e.g., schizophrenia, major depression, history of psychiatric disorders); (3) diagnosis of obstructive sleep apnoea; (4) past surgery in the abdominal area; (5) any use of medication. Each fasting participant was subjected to a clinical and anthropometric assessment, as well as abdominal ultrasonography for VAT and SAT measurements during the same morning, and they were asked to fill two complete validated questionnaires, one assessing the chronotype and one the adherence to the traditional MD. Of the 479 patients initially recruited, we excluded those with VAT over 12 cm, due to the difficulty of measurement (23), and those lacking one of the variables of interest (40). The final dataset included a total of 416 subjects. The study complied with the principles established by the Declaration of Helsinki, and written informed consent was obtained by each subject. The Ethical Committee of the University of Milan (n. 6/2019) approved the study procedures.

### 2.2. Clinical and Anthropometrical Assessment

An accurate medical interview was carried out, along with the collection of medical condition self-reported diagnosis and information regarding current drug therapies. A structured interview was employed to investigate physical activity levels. Subjects engaging in any structured physical exercise for >2 h/week were deemed as physically active. The conventional criteria and measuring procedures set out by Lohman et al. [23] were used to collect the anthropometric measurements. Body weight (BW, kg) and body height (BH, cm) were measured to the nearest 100 g and 0.1 cm, respectively. A column scale (Seca 700 balance, Seca Corporation, Hamburg, Germany) was employed to measure BW, the measurement being carried out after bladder emptying with participants wearing only light underwear. A vertical stadiometer was employed to measure BH. The calculation of the body mass index (BMI) was performed using the following formula: BW (kg)/BH^2^ (m^2^). Obesity classification followed the WHO guidelines [24]. Waist circumference was determined midway between the lower rib margin and the superior anterior iliac spine using a horizontally applied non-stretch tape, and it was measured to the nearest 0.5 cm.

### 2.3. Abdominal Ultrasonography

VAT and SAT were measured by the same operator following a validated standardized protocol [25] and using a Logiq 3 Pro system for abdominal ultrasonography equipped with a 3.5 MHz convex-array probe and a 7.5 MHz linear probe (GE Healthcare, Milwaukee, WI, USA). VAT was measured as the distance between the posterior surface of the rectus abdominis muscle and the anterior wall of the aorta at the level of linea alba, and SAT was measured as the distance between the external face of the rectus abdominis muscle and the epidermis. Both VAT and SAT were determined three times, one centimetre above the umbilicus, and a mean measurement was computed. The intra-operator coefficient of variation for repeated VAT and SAT measurements in our laboratory is 0.8%. VAT to SAT ratio was also calculated.

### 2.4. Adherence to the Mediterranean Diet

Adherence to the traditional MD was assessed using a validated 14-item questionnaire [26]. The guidelines set out by the Prevención con Dieta Mediterránea (PREDIMED) study group (www.predimed.es) with some adaptation already employed in previous studies were used to obtain the Mediterranean score (MED score) [19,25,26,27,28]. One point was attributed for each of the following: (1) olive oil as the main cooking fat; (2) olive oil ≥ 4 tablespoons/day; (3) vegetables > 2 servings/day (or *≥* 1 portion raw or salad); (4) fruit ≥ 3 servings/day; (5) red or processed meat < 1 serving/day; (6) butter or cream or margarine < 1/day; (7) sugar-sweetened beverages < 1/day; (8) wine ≥ 3 glasses/week; (9) legumes *≥* 3 servings/week; (10) fish/seafood *≥* 3 servings/week; (11) commercial sweets and confectionery < 3/week; (12) nuts ≥ 1/week; (13) white more than red meats (yes); and (14) use of sofrito ≥ 2/week. Participants with a MED score ≥ 9 points were considered as complying with a dietary pattern in accordance with the MD [26].

### 2.5. Chronotype

A shortened 5-item version of the standard 19-item Morningness-Eveningness Questionnaire (MEQ) [16] was used to assess chronotype. Due to its good stability, reliability coefficient range, as well as translation and validation in different languages [27] MEQ is one of the most widely used questionnaires for chronotype determination. This shortened version (rMEQ) accounts for 83% of the original MEQ total variance [11,28] These five items assess sleeping and waking hours, peak time, morning alertness, and self-assessment of chronotype. Each item corresponds to a score: the sum of the scores allows to obtain the final rMEQ score, which varies from < 12 (extreme E-Types, more active in the last part of the day) to > 17 (extreme M-Types, who feel wide awake and fresh in the morning). Intermediate scores were associated with N-Types (12–17 points), who have a more flexible sleep period. In the analysis, the sum score was used either as a continuous variable or as a categorical variable by dividing said score into tertiles. Eveningness was represented by the lowest tertile, intermediate preference was represented by the mid tertile, and morningness was represented by the highest tertile.

### 2.6. Statistical Analysis

All continuous variables followed a Gaussian distribution; therefore, they are presented as mean ± standard deviation (SD). Discrete variables are reported as frequencies and proportions. Descriptive characteristics were compared among chronotype groups using one-way ANOVA with Bonferroni post hoc test for numeric variables. The Χ^2^ test was used to compare categorical variables.

The association of rMEQ score with the continuous outcomes of interest (WC, VAT, SAT, VAT/SAT ratio) was evaluated using pre-specified linear regression models and robust confidence intervals, including sex (discrete, 0 = female; 1 = male), age (continuous, years), BMI (continuous, kg/m^2^), and MED score (continuous, units) as confounders. A *p*-value < 0.05 was considered statistically significant. Statistical analysis was performed using IBM SPSS Statistics software version 26.0 for Windows (IBM, Armonk, NY, USA).

## 3. Results

69.5% of patients were women and 30.5% were men. The general characteristics of the study subjects are reported in Table 1. In our population, 30.8% were M-Types, 10.4% were E-Types, and the remaining were N-Types. rMEQ score was 19 ± 1 in M-Types, 15 ± 2 in N-Types, and 10 ± 1 in E-Types. No difference in BMI and abdominal obesity and fat distribution occurred among chronotypes. However, MED score was significantly higher in M-Types compared to E-Types (7 ± 2 vs. 6 ± 2; *p* < 0.05).

Table 2 shows the prevalence of sex, normal weight, overweight and obesity, and adherence to MD in the different chronotypes.

No differences in the prevalence of sex, BMI classes, and physical activity level occurred among groups. Only adherence to MD resulted significantly lower in the E-Types compared to M-Types.

Table 3 reports the contribution of rMEQ score to WC, SAT, VAT, and VAT/SAT, considering sex, age, BMI, and MED score.

Being male, having a higher age and a greater BMI increase WC, visceral compartment and VAT:SAT ratio. In addition, WC decreases with increasing MED score and rMEQ score. In particular, visceral fat decreases with increasing morningness, indicating a lower abdominal visceral fat for every 1 point of rMEQ score, independently of sex, age, BMI and adherence to MD.

## 4. Discussion

In this study, we investigated the contribution of chronotype to abdominal obesity and its fat distribution within a sample of white European subjects, considering sex, age, BMI, but also lifestyle factors such as physical activity and adherence to MD. Overall, we found that rMEQ score was inversely associated with both waist circumference and visceral fat. Moreover, E-Types reported a lower adherence to a balanced MD.

Abdominal obesity is an important driving force behind the progression of multiple cardiometabolic diseases [3], especially at visceral level, for its endocrine and paracrine functions [8,29]. A lifestyle modification, particularly one related to the MD, remains the treatment of choice [10], though with poor and short-term results in any case [1,30]. For these reasons, finding as many abdominal obesity determinants as possible may improve the personalization of nutritional approaches and their effectiveness. 

In this scenario, chrononutrition is an emerging research field [15]: timing, frequency, and regularity of daily life and relative fasting-feeding cycles seem to play a significant role in the relationship between diet and cardiometabolic health [31]. Multiple oscillators distributed throughout the body along with a central pacemaker located within the suprachiasmatic nucleus of the hypothalamus coordinating the whole structure are the main components constituting the mammalian circadian system [32]. A number of genes, commonly termed clock genes, represent the molecular mechanism underlying the circadian function [14]. The coordinates of circadian behaviour and organism physiology are known to regulate the effectiveness of a core function of the molecular clock machinery, referred to as circadian output [33]. Despite the role of circadian system in controlling human rhythms, diet itself can modify homeostasis and change the biological rhythms [32,34]. The biological rhythms in humans present interindividual differences that determine their chronotype or circadian typology, representing the tendency to show morning or evening preferences [35]. Growing evidence shows that E-Types tend to eat fewer, larger, and unwholesome meals [15,20], probably due to a poor sleep quality [12,18] or lower self-control levels [11,12,13,15], and are at increased risk for cardiovascular diseases [15]. Nevertheless, very few studies have been focused on this field. 

In 2015, Yu et al. [12] revealed for the first time an association between later chronotype and sarcopenia, metabolic disorders, and abdominal obesity. They found that E-Type typology was independently associated with sarcopenia, diabetes, and metabolic syndrome, regardless of lifestyle, supporting the importance of circadian rhythms in the metabolic regulation [12]. No evidence of association between chronotype and abdominal obesity and visceral fat was found; moreover, the interplay of diet in this association was not investigated. This latter aspect was examined for the first time by Finnish researchers, studying the interrelation between chronotype, the Baltic Sea diet and obesity [11]. They showed that, despite E-types’ lower adherence to a balanced diet, chronotype would not change the association between a balanced diet and obesity. Interestingly, only one recent study examined the association of chronotype with MD and obesity in young adults (18–25 years) [22], showing that E-Types with greater social jet lag had lower adherence to the MD and a higher BMI. Nonetheless, they did not investigate abdominal obesity and its fat distribution. 

Our results showed no differences in WC and VAT between chronotypes, confirming the results of Yu et al. [12]. By contrast, we found a negative association between rMEQ score and WC and VAT, showing a lower WC and VAT (−0.2 cm and −0.06 cm, respectively) for every 1 point of rMEQ score, indicating that E-Types have +2 cm of WC and +0.5 cm of VAT compared to M-Types, independently of sex, age, BMI, physical activity, and adherence to MD. These conflicting results compared to Yu et al. [12] could be explained by our consideration of the contribution of a balanced diet, such as MD, in the association. 

As in the Finnish study [11], we showed that E-Types had an overall lower adherence to a balanced diet as compared to the other chronotypes, as also shown in young adults by Zerón-Rugerio et al. [22]; nonetheless, as opposed to Maukonen et al. [11], we confirmed a negative association between chronotype and WC, independently of confounding factors. 

To the best of our knowledge, this is the first study performed on adults to examine the contribution of chronotype to abdominal obesity and its fat distribution, considering the adherence to the MD. Our study makes a significant contribution to the emerging literature on the interaction between chronotype and cardiometabolic health, and to the role of diet in this interplay, but a number of potential limitations need to be addressed. The sample size is relatively small; it is limited especially by the unrepresentativeness of males; therefore more studies are needed. Another limitation is that abdominal fat was not measured by a gold standard method, such as computed tomography scans or magnetic resonance imaging, but using ultrasonography. However, all ultrasound measurements were taken by the same physician, who has an extended experience in the use of ultrasonography [5,10,25]; furthermore, evidence has shown a good correlation between this field method versus the gold standards [25]. Lastly, the cross-sectional design does not permit identification of cause and effect or temporality of association. So, further studies are needed to understand the relation between circadian typology, abdominal obesity and diet.

## 5. Conclusions

In conclusion, our results have supported the growing literature on the health-impairing eating habits of E-Types compared to other chronotypes. Our novel finding was the significant contribution of chronotype to abdominal obesity, particularly to the visceral component, showing that eveningness is a potential determinant of abdominal obesity and its visceral component. These results underline the potential implications of the individual circadian typology on abdominal fat distribution, and the importance of considering chronotype in the implementation of a tailor-made dietary treatment for the prevention and treatment of abdominal obesity. 

## Figures and Tables

**Table 1 nutrients-12-00592-t001:** General characteristics of the study subjects.

	Chronotypes							
	M-Types		N-Types		M-Types		Total	
	*N* = 135 (32.5%)		*N* = 243 (58.4%)		*N* = 38 (9.1%)		*N* = 416	
	Mean	sd	Mean	sd	Mean	sd	Mean	sd
Age (years)	51	12	48	13	51	13	50	13
BMI (kg/m^2^)	29.7	5.6	29.1	6.1	29.4	6.1	29.6	6.3
Abdominal fat distribution								
WC (cm)	98.4	13.2	97.8	14.5	99.6	13.5	99	14.9
SAT (cm)	2.6	1.3	2.5	1.1	2.5	1.3	2.5	1.2
VAT (cm)	5.1	2.3	5.1	2.5	5.2	2.9	5.3	2.7
VAT/SAT	2.5	2.2	2.4	1.6	3.2	4.3	2.6	2.4
Mediterranean Diet								
MED score	7	1	7	2	6 *****	2	7	2

* *p* < 0.050 vs. M-Type. BMI = body mass index; SAT = subcutaneous adipose tissue; VAT = visceral adipose tissue; WC = waist circumference; MD = Mediterranean diet; MED score = Mediterranean diet questionnaire score; E-Types = evening type; M-Types = morning type; N-Types = neither-Types.

**Table 2 nutrients-12-00592-t002:** Prevalence of sex, BMI classes, and adherence to MD among chronotypes.

	Chronotypes							
	M-Types		N-Types		E-Types		Total	
	*N* = 135 (32.5%)		*N* =243 (58.4%)		*N* = 38 (9.1%)		*N* = 416	
	*n*	%	*n*	%	*n*	%	*n*	%
Sex								
Male	39	28.9	80	32.9	8	21.1	127	30.5
Female	96	71.1	163	67.1	30	78.9	289	69.5
Total	135	100.0	243	100.0	38	100.0	416	100.0
BMI classes								
Normal weight	23	17.1	69	28.4	10	26.3	102	24.5
Overweight	52	38.5	87	35.8	12	31.6	151	36.3
Obese	60	44.4	87	35.8	16	42.1	163	39.2
Total	135	100.0	243	100.0	38	100.0	416	100.0
Physical Activity								
No	103	76.3	199	81.9	33	86.8	335	80.5
Yes	32	23.7	44	18.1	5	13.2	81	19.5
Total	135	100.0	243	100.0	38	100.0	416	100.0
Adherence to MD								
No	112	83.0	193	79.4	35	92.1	340	81.7
Yes	23	17.0	50	20.6	3	7.9^*^	76	18.3
Total	135	100.0	243	100.0	38	100.0	416	100.0

* *p* < 0.050 vs. M-Types, E-Types = evening Types; M-Types = morning Types; N-Types = neither-Types; MD = Mediterranean diet.

**Table 3 nutrients-12-00592-t003:** Association between chronotype and abdominal obesity.

	WC (cm)	SAT (cm)	VAT (cm)	VAT/SAT
Sex				
male	ref.	ref.	ref.	ref.
female	−8.66 ***	0.26 *	−1.99 ***	−1.60 ***
	[−9.67, −7.66]	[0.00, 0.51]	[−2.39, −1.60]	[−2.21, −0.99]
Age (years)	0.15 ***	−0.16 ***	0.04 ***	0.04 ***
	[0.11, 0.19]	[−0.02, −0.01]	[0.03, 0.06]	[0.02, 0.06]
BMI (kg/m^2^)	2.07 ***	0.10 ***	0.26 ***	0.01
	[1.96, 2.17]	[0.09, 0.12]	[0.23, 0.29]	[−0.20, 0.46]
Physical activity				
No	ref.	ref.	ref.	ref.
Yes	0.91	0.16	0.21	−0.06
	[−0.37, 2.20]	[−0.08, 0.41]	[−0.16, 0.58]	[−0.54, 0.41]
MED score	−0.31 *	−0.03	0.01	0.04
	[−0.60, −0.01]	[−0.08, 0.02]	[−0.08, 0.11]	[−0.08, 0.16]
rMEQ score	−0.19 **	−0.03	−0.06 *	−0.05
	[−0.33, −0.06]	[−0.01, −0.06]	[−0.11, −0.01]	[−0.13, −0.03]

* *p* < 0.050; ** *p* < 0.010; *** *p* < 0.001, BMI = body mass index; MED score = Mediterranean diet questionnaire score; rMEQ score = reduced morningness-eveningness questionnaire score; SAT = subcutaneous adipose tissue; VAT = visceral adipose tissue; WC = waist circumference.

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
