# Peer review of "Is Abdominal Fat Distribution Associated with Chronotype in Adults Independently of Lifestyle Factors?"

_nutrients, 2020, doi:10.3390/nu12030592_

Round 1
Reviewer 1 Report
This paper has some important limitations, notably the small sample size and less than optimal methods of obtaining the data, but the authors are well aware of these limitations and state them fairly in the discussion. It is perhaps not surprising that the effects observed are relatively small and of limited statistical significance. Nevertheless I agree with the authors that the paper makes a significant contribution to what is currently a sparsely reported field of investigation. The results are of some interest to researchers in this field but I would recommend some minor textual changes to avoid overstating what has been shown in relation to health outcomes. For example in the last line of the abstract, and in the conclusions it is stated that the results underline the role of individual circadian typology on cardiometabolic health. This is not strictly true as no measurements of cardiometabolic health have been obtained or reported. Please rephrase to make it clear that the results that are reported only have some potential or theoretical implications for health, which remain to be explored. I also find the multiplicity of abbreviations makes the results section of the paper a little difficult to follow. Please amend the text to make it a little more clear how the rMEQ score is calculated, and how the scores obtained relate to actual subject behaviour.
Author Response
This paper has some important limitations, notably the small sample size and less than optimal methods of obtaining the data, but the authors are well aware of these limitations and state them fairly in the discussion. It is perhaps not surprising that the effects observed are relatively small and of limited statistical significance. Nevertheless I agree with the authors that the paper makes a significant contribution to what is currently a sparsely reported field of investigation.
We thank the reviewer for the considerations.
The results are of some interest to researchers in this field but I would recommend some minor textual changes to avoid overstating what has been shown in relation to health outcomes. For example in the last line of the abstract, and in the conclusions it is stated that the results underline the role of individual circadian typology on cardiometabolic health. This is not strictly true as no measurements of cardiometabolic health have been obtained or reported. Please rephrase to make it clear that the results that are reported only have some potential or theoretical implications for health, which remain to be explored.
We agree with the reviewer. We have rephrased both abstract (lines 40,41) and conclusions (lines: 265-268):
“In conclusion, these results showed that chronotype is independently associated with abdominal obesity and visceral fat, underlining the potential implications of the individual circadian typology on abdominal obesity”.
“These results underline the potential implications of the individual circadian typology on abdominal fat distribution, and the importance of considering chronotype in the implementation of a tailor-made dietary treatment for the prevention and treatment of abdominal obesity”.
- I also find the multiplicity of abbreviations makes the results section of the paper a little difficult to follow. Please amend the text to make it a little more clear
We thank the reviewer for the suggestion. We have rephrased the results section at lines 193-196:
“Being male, having a higher age and a greater BMI increase WC, visceral compartment and VAT:SAT ratio. In addition, WC decreases with increasing MED score and rMEQ score. In particular, visceral fat decreases with increasing morningness, indicating a lower abdominal visceral fat for every 1 point of rMEQ score, independently of sex, age, BMI and adherence to MD”.
- how the rMEQ score is calculated, and how the scores obtained relate to actual subject behaviour.
We thank the reviewer for the suggestion. We added the following sentence at lines 156-158:
“Each item corresponds to a score: the sum of the scores allows to obtain the final rMEQ score, which varies from < 12 (extreme E-Types, more active in the last part of the day) to > 17 (extreme M-Types, who feel wide awake and fresh in the morning). Intermediate scores were associated with N-Types (12-17 points), who have a more flexible sleep period”.

Reviewer 2 Report
In a cross-sectional observational study, the authors have examined the relationship between chronotypes and diet and abdominal obesity in a group of white European subjects. Their data show that eveningness was associated with greater abdominal and visceral adiposity.
The study is done well and the data are clearly presented.
The weakness include the crossesctional nature of the study, a single observation and the much smaller sample size of the E-Type.
The manuscript requires moderate rewrite for language. The authors should avoid the non-scientific terms like healthy and unhealthy,caucasian (is no longer an accepted term) etc.
Author Response
- In a cross-sectional observational study, the authors have examined the relationship between chronotypes and diet and abdominal obesity in a group of white European subjects. Their data show that eveningness was associated with greater abdominal and visceral adiposity. The study is done well and the data are clearly presented. The weakness include the crossesctional nature of the study, a single observation and the much smaller sample size of the E-Type.
We thank the reviewer for the appreciation.
- The manuscript requires moderate rewrite for language. The authors should avoid the non-scientific terms like healthy and unhealthy,caucasian (is no longer an accepted term) etc.
We thank the reviewer for the suggestion. Another native English speaker reviewed the manuscript and we deleted all the terms mentioned.
